# Intercomparison of Pandora Stratospheric NO$_2$ Slant Column Product with the NDACC-Certified M07 Spectrometer in Lauder, New Zealand

T. N. Knepp[1,2], R. Querel[3], P. Johnston[3], L. Thomason[2], D. Flittner[2], and J. M. Zawodny[2]

[1]Science Systems and Applications Inc., Hampton, VA, 23666 United States
[2]NASA Langley Research Center, Hampton, VA, 23681 United States
[3]National Institute of Water and Atmospheric Research, Lauder, Central Otago, New Zealand

*Correspondence to:* T. N. Knepp
(travis.n.knepp@nasa.gov)

**Abstract.** In September 2014, a Pandora multi-spectral photometer operated by the SAGE-III project was sent to Lauder, New Zealand to operate side-by-side with the National Institute of Water and Atmospheric Research's (NIWA) Network for Detection of Atmospheric Composition Change (NDACC) certified zenith slant column NO$_2$ instrument to allow intercomparison between the two instruments, and for evaluation of the Pandora unit as a potential SAGE-III validation tool for stratospheric NO$_2$. This intercomparison spanned a full year, from September 2014 – September 2015. Both datasets were produced using their respective native algorithms using a common reference spectrum (i.e. 12:00 on 26-February 2015). Throughout the entire deployment period both instruments operated in a zenith-only observation configuration. Though conversion from slant column density (SCD) to vertical-column density (VCD) is routine (by application of an air mass factor), we limit the current analysis to SCD only. This omission is beneficial in that it provides an intercomparison based on similar modes of operation for the two instruments and the retrieval algorithms as opposed to introducing an air mass factor dependence in the intercomparison as well. It was observed that the current hardware configurations and retrieval algorithms are in good agreement (R > 0.95). The detailed results of this investigation are presented herein.

## 1   Introduction

The Stratospheric Aerosol and Gas Experiment (SAGE) missions have provided a legacy of high-quality solar occultation measurements for vertically profiling stratospheric O$_3$ and UV/VIS/NIR aerosol extinction coefficients from the upper troposphere into the mesosphere for more than three decades (Chu and McCormick, 1979, 1986; Damadeo et al., 2013; Cisewski et al., 2014). These observations have formed a crucial component for understanding ozone trends, and how stratospheric chemistry and aerosol influence ozone mixing ratios and climate. An updated version of the SAGE instrument (hereafter referred to as SAGE-III) was integrated into the International Space Station (ISS) in March 2017 with routine observations starting in April. The SAGE-III project will focus on reassessing the state of stratospheric O$_3$ recovery and provide requisite aerosol observations for climate and ozone models. To this end, the standard data products for this mission are aerosol extinction coefficients,

aerosol optical depth, $O_3$, $H_2O$, and $NO_2$ mixing ratios. For an overview of the instrument and products see Cisewski et al. (2014).

As with any new instrument, a significant post-launch activity is planned to validate the accuracy and precision of the data products, and provide validated datasets to end users. While the key SAGE-III species measurements are validated using well-known and characterized instruments, one important product remains difficult to validate: $NO_2$. $NO_2$ is important due to its role in partitioning stratospheric odd hydrogen, providing a chemical pathway for conversion of ozone-destroying species to their reservoir forms (e.g. halogen species as discussed by Wennberg et al. (1994)), and may be responsible for up to 70 % of stratospheric ozone loss (Crutzen, 1970; WMO, 1985; Seinfeld and Pandas, 1998; Chartrand et al., 1999; Portmann et al., 1999). The quality of the $NO_2$ retrievals also impacts the quality of short wavelength aerosol extinction coefficient measurements as well as, to a lesser extent, ozone.

Observations are made over a large range of latitudes depending on season and the details of the orbit but only at two latitudes on a given day (where the spacecraft crosses the terminator or (given the question) each sunrise and sunset encountered by the spacecraft (one of each per orbit). Due to the unique viewing geometry of SAGE-II and the rapid variability of $NO_2$ across the solar terminator, $NO_2$ measurements from previous SAGE missions (SAGE-II and SAGE-III/Meteor) proved to be challenging. For SAGE-III/Meteor, $NO_2$ is often validated using measurements from other space-based instruments that generally do not fully match the SAGE viewing geometry, location and/or time. While a chemistry model can correct for some of these differences, generally these comparisons leave significant questions regarding the $NO_2$ data quality. Given the variability and relative sparsity of observations, Pandora provides a unique capacity to be carried to a measurement location rather than only providing data when an observation occurs near a fixed site. This enables observations from places that are challenging for the SAGE instrument particularly where strong gradients across the tropopause may occur (like the tropics) or other observations of opportunity (i.e. various field campaigns).

An alternative method that provides some corroboration to the SAGE-III measurement quality is comparison with ground-based Differential Optical Absorption Spectroscopy (DOAS, e.g. Platt and Stutz, 2008) or Fourier Transform Spectroscopy (FTS, e.g. Wang et al., 2010) measurements of the column $NO_2$ using zenith-looking instruments that measure scattered light across the ultra-violet and visible wavelengths. These observations can be used to infer, among other species, column $NO_2$ as a function of solar zenith angle (SZA). Zenith-viewing observations when SZA $\approx 90°$ are analogous to solar occultation measurements of $NO_2$. However, observation of stratospheric $NO_2$ is challenging at many locations due to the high levels of tropospheric $NO_2$ from human-derived sources. Therefore, measurement sites in locations that are considered "background-level" are advantageous.

The National Institute of Water and Atmospheric Research (NIWA) Lauder, New Zealand site provides the required tropospheric background-level conditions for observation of stratospheric $NO_2$. The Lauder group has a long history of providing high-quality observations for stratospheric $NO_2$ and $O_3$ (McKenzie and Johnston, 1982; McKenzie et al., 1992). Data collected at the Lauder site have been used to infer data quality for SAGE II $NO_2$ and were used to identify and help correct a time-dependent error in those observations (Damadeo et al., 2013). For the new SAGE-III mission, observations by the NIWA instrument will be useful for understanding $NO_2$ data quality. However, since the challenges of making space-based measure-

ments is often latitude dependent, a single site will not provide all the corroborative data needed to make a robust assessment of data quality. As a result, the SAGE-III group has acquired a Pandora unit (Herman et al., 2009; Tzortziou et al., 2015) with the hope of using it as a portable system for providing corroborative data that can be deployed at sites of opportunity, for instance low latitudes, throughout the SAGE-III/ISS mission. To date, Pandoras have not established a record for measuring $NO_2$ where the column is dominated by the stratosphere rather than a polluted troposphere so an evaluation of the capabilities of this instrument in this regard is necessary. Herein, we report the results of a comparison of observations by a NIWA owned/operated instrument and the SAGE-III Pandora unit when operated side-by-side between September 2014 and September 2015 at the NIWA facility in Lauder, New Zealand.

## 2 Instrumentation

### 2.1 LaRC Pandora

Pandora is a sun-viewing spectrometer that was initially developed for validation of the Ozone Monitoring Instrument (OMI) aboard the Aura satellite (Herman et al., 2009; Lamsal et al., 2014; Tzortziou et al., 2015), and has proven to be sensitive to fluctuations in boundary layer $NO_2$ over short time periods (Knepp et al., 2015). Due to Pandora's potential for retrieving stratospheric gas column densities (i.e. operating in zenith orientation during twilight hours) it has been evaluated as a potential validation instrument for the SAGE-III mission.

A detailed description of the instrument has been provided by Herman et al. (2009). Briefly, the Pandora model used in the current study consisted of: 1. an optical head (mounted on a two-axis tracker capable of moving through 360$^{\circ}$ azimuth and 90$^{\circ}$ zenith) containing filter wheels for controlling polarization and radiant flux; 2. a single-strand, multi-mode fiber-optic cable with 400 $\mu$m core diameter and numerical aperture of 0.22 to transmit photons to the spectrometer; 3. a temperature stabilised Avantes spectrometer (model number ULS2048x64, 280 – 525 nm) with a 50 $\mu$m slit, focal length of 75 mm, and resolution on the order of 0.6 nm; 4. laptop computer for instrument control and data logging. The improved optics and spectrometer of this model enabled the instrument to record solar spectra from lunar reflectance and scattered radiation, which has spurred investigation regarding its ability to accurately estimate the slant-column density (SCD) of stratospheric species from twilight spectra.

The Pandora retrieval algorithm was previously described in Herman et al. (2009), Tzortziou et al. (2015) and Cede (2017), with relevant cross-section details presented in Table 1. Briefly, spectral fitting is performed using laboratory-measured absorption cross sections and implement shift-squeeze functions to fit the observed spectra with the solar reference spectrum's Fraunhofer line structure (for zenith observations an instrument-observed solar-reference spectrum was used from the spectrum recorded at 12:00 (local time) on 26-February 2015), with a fourth-order polynomial applied for removal of aerosol and Rayleigh scattering effects.

Though Pandora was developed to operate in a Sun-tracking mode and has undergone numerous revisions to allow data collection in sky (i.e. scattered irradiance for elevation scans) and moon observation modes, the instrument's capability of making accurate twilight observations remained unknown. Part of the motivation of the current study was to evaluate Pandora's ability

|  | Instrument | Setting |
|---|---|---|
| $O_3$ Cross Section | Pandora | Daumont et al. (1992); Malicet et al. (1995) (225 K, 300 – 330 nm) |
|  | M07 | Brion et al. (1993) (218 K, 428 – 469 nm) |
| $NO_2$ Cross Section | Pandora | Vandaele et al. (1998) (220 K, 400 – 485 nm) |
|  | M07 | Vandaele et al. (1998) (220 K, 428 – 469 nm) |
| $O_4$ Cross Section | Pandora | Smith et al. (2001) (262 K, 400 – 454 nm) |
|  | M07 | Thalman and Volkamer (2013) (262 K, 428 – 469 nm) |
| Ring | Pandora | Thuillier et al. (2004) |
|  | M07 | NDACC recommended pseudo cross section Chance and Spurr (1997) |
| Polynomial Order | Pandora | $4^{th}$ |
|  | M07 | $3^{rd}$ |

**Table 1.** Relevant retrieval details for the two instruments under study.

to make reliable twilight observations, thereby demonstrating its applicability to SAGE-III validation. To this end, Pandora only operated in the zenith-observation mode to allow direct intercomparison with the zenith-oriented NIWA instrument.

### 2.2 NIWA spectrometer

The NIWA instrument (M07) is a zenith-oriented instrument used for measuring stratospheric slant column $NO_2$. M07 is the current instrument contributing to the continuous time series of stratospheric $NO_2$ from Lauder that started in 1980, and is part of the Network for Detection of Atmospheric Composition Change (Hofmann et al., 1995; Roscoe et al., 1999). The NDACC-certified M07 instrument has been described previously (McKenzie and Johnston, 1982; McKenzie et al., 1992; Hofmann et al., 1995) as has the STRATO retrieval software that was build in-house (Peters et al., 2017). Briefly, M07 is a Czerny-Turner monochromator (320 mm focal length, ≈0.8 nm resolution, F/5 entrance field of view, 1 mm wide slit) with a bi-alkali photocathode photomultiplier detector. The scanning mechanism was modified to provide fast scanning with a long lifetime and smooth wavelength motion. The instrument is mounted in a temperature controlled cabinet on a rotating table following the line of the sun-zenith plane and a Glan-Thompson polariser is used in front of the entrance slit to provide polarised zenith measurements. Similar to the Pandora, the cross sections used for retrievals are listed in Table 1.

### 2.3 Uncertainties

Within the scope of the current work, the dominant source of uncertainty for each instrument, during twilight conditions, was statistical uncertainty due to limited light throughput. In this regard, Pandora is inferior to M07 (*vide supra*) as it was initially designed for direct-sun observations. Other sources of uncertainty that have less impact within the current analysis are slant-column amount in the chosen reference spectrum, fitting settings such as $NO_2$ temperature, and retrieval technique.

## 3 Mode of operation and location

The Pandora unit was deployed to the NIWA station in Lauder, Central Otago, New Zealand (45.038 S, 169.68 E, 370 m ASL) to run side-by-side with the NIWA-operated M07 spectrometer. Both instruments performed retrievals using a common reference spectrum (collected on 26-February, 2015 12:00 local time) as observed by the respective instruments. It is worth noting that, other than the Pandora's fixed zenith observation state, both instruments were operated in their normal states, not in a customized operation mode, and both used their standard retrieval algorithms.

New Zealand is generally an atmospherically clean environment, with pollution levels that can be considered as background level (e.g. approximately an order of magnitude below urban centers in the continental United States). As a point of reference, $NO_2$ retrievals (VCD) over New Zealand and the continental United States from the OMI (Level 3, version 3 algorithm) are presented on the same scale in Figs. 1 and 2. It is observed that aside from some western states (e.g. Nevada and Oregon), the U.S. rarely experiences similarly clean conditions as New Zealand. Furthermore, Fig. 2 displays a downward trend in overall column $NO_2$ over the Chemistry and Physics of the Atmospheric Boundary Layer Experiment (CAPABLE) station located at NASA's Langley Research Center in Hampton, VA (37.103 N, -76.387 E, 5 m ASL) that is being driven by a decreasing tropospheric column, while $NO_2$ over Lauder has remained consistent since 2005. There is no corresponding change in stratospheric $NO_2$ for either site.

Statistics describing the variability in $NO_2$ over both sites were broken into three categories (total-column, stratospheric and tropospheric contributions) and are presented in Table 2. The statistics presented in Table 2 are similar for both sites when scaled according to corresponding column density (i.e. despite the total-column standard deviation being significantly different for both sites the relative error ($\sigma/\overline{x}$) remains similar). Despite these similarities, the tropospheric variability remains different for the two sites indicating a higher degree of variability over the continental US. The tropospheric contribution and variability remains significantly higher over CAPABLE as compared to Lauder with approximately 55.7 % of the $NO_2$ column residing in the troposphere over the CAPABLE site, and only 13.5 % over Lauder. These differences are driven by the ubiquity of local sources in the eastern United States as compared to central New Zealand.

Figures 1 and 2 and Table 2 demonstrate that not only is the $NO_2$ column significantly higher over the continental US, but the Lauder column is dominated by the stratospheric contribution. One effect of differing source strengths is seen in Fig. 2. Being in different hemispheres, the two sites should be approximately six months offset in their seasonal cycle, though the total-column time series shows the two sites are in phase. However, the stratospheric contribution for the two sites (panel (b)) remained approximately six months out of phase as expected. This can be explained by the difference in tropospheric and stratospheric chemistry. Under normal, moderately-polluted conditions, tropospheric chemistry is sufficiently perturbed to force the tropospheric $NO_2$ column density out of phase with the stratosphere. This shows up in the data as an approximately six month offset. Since the northern hemisphere site is dominated by tropospheric $NO_2$ (as shown in Fig. 2 and Table 2), and the southern hemisphere site has significantly less tropospheric contribution, the two sites are in phase with one another in the total-column panel. Nitrogen dioxide SCDs from the ground instruments (Fig. 2, panel (b)) show the surface instruments are accurately detecting the stratospheric seasonal cycle (i.e. are in phase with the OMI stratospheric column over Lauder).

Since Lauder provides a clean, background-level, environment with few local or regional anthropogenic emission sources, it provides ideal conditions for observation of stratospheric species and evaluation of the Pandora system for detecting stratospheric $NO_2$ and as a possible validation tool for current and upcoming satellite missions that focus on stratospheric chemistry.

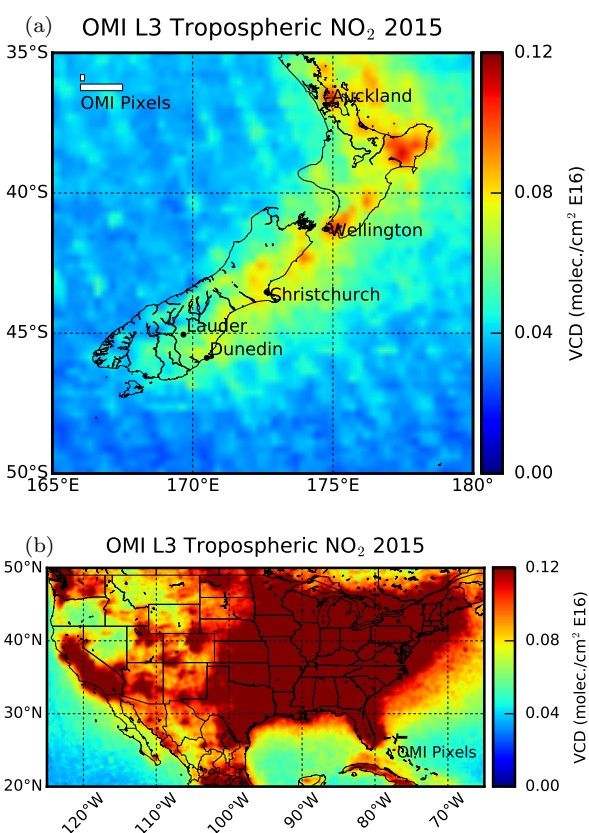

**Figure 1.** Annual average for OMI $NO_2$ (L3, v3.0) maps over New Zealand and North America. OMI pixel sizes (nadir and swath edge) are represented by the white boxes within each panel. For comparison purposes, both plots were put on the same color scale.

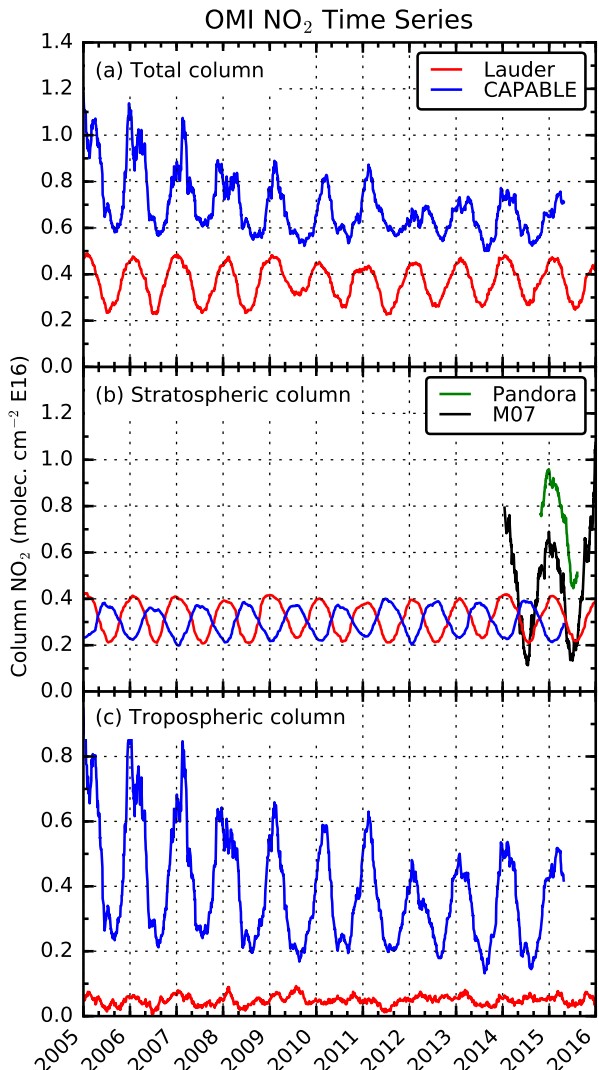

**Figure 2.** Time series plots for total- (a), stratospheric- (b), and tropospheric-column (c) NO₂ data products from OMI (L3, v3.0, VCD) over Lauder (red) and CAPABLE (blue). OMI data were filtered to remove cloud fractions greater than 20 % and overpasses greater than 50 km from the site. A seven-day normally-weighted rolling mean was applied to smooth the plots and remove higher-frequency fluctuations. Pandora (green) and M07 (black) data presented in panel (b) are slant-column densities.

| Parameter | Lauder | CAPABLE |
|---|---|---|
| Total-column ($\overline{x}$) | 0.37 | 0.70 |
| Total-column ($\sigma$) | 0.08 | 0.14 |
| Total-column ($\sigma/\overline{x}$) | 0.21 | 0.19 |
| Stratosphere ($\overline{x}$) | 0.32 | 0.31 |
| Stratosphere ($\sigma$) | 0.07 | 0.06 |
| Stratosphere ($\sigma/\overline{x}$) | 0.21 | 0.19 |
| Stratospheric Fraction (%) | 86.5 | 44.3 |
| Troposphere ($\overline{x}$) | 0.05 | 0.39 |
| Troposphere ($\sigma$) | 0.01 | 0.17 |
| Troposphere ($\sigma/\overline{x}$) | 0.29 | 0.45 |
| Tropospheric Fraction (%) | 13.5 | 55.7 |

**Table 2.** Statistics regarding stratospheric and tropospheric contribution and variability of $NO_2$ (VCD) as observed by OMI. All units are molec cm$^{-2}$ E16.

## 4 Intercomparison

Pandora and M07 data were filtered to remove points where the retrieval uncertainty was greater than 10 % of the retrieved value followed by resampling to five-minute means to allow direct, temporally-aligned, intercomparisons. Unless otherwise noted, all intercomparisons and analyses were carried out using 5 min averaged data.

### 4.1 Aggregate analysis

An aggregate analysis was performed on the resampled data by binning the SCD according to SZA for a visual evaluation of the correlations as seen in Fig. 3. It is observed that the correlation is generally poor during pre-sunrise/twilight hours (i.e. when SZA>92.5°), but improves with decreasing SZA where it peaks at 80-85° (Table 2). At 95 % confidence, all R$^2$ values in Table 2 are significantly different except when comparing the 87.5-90.0° bin with either the 85.0-87.5° or 80.0-85.0° bins; however, the 80.0-85.0° and 87.5-90.0° bins remain statistically different. Within each panel of Fig. 3 the data are color coded to correspond to the SZA range within each sub-panel and provide insight into how the short-term change in SZA influenced agreement. As an example, in panel (a) it is observed that data collected at higher SZA (red-shaded points) were further from the one-to-one line than data collected at lower SZA (blue-shaded points). Analogously, it is observed in panels (e-g) that as SZA decreased, so too did the degree of correlation. Therefore, we can conclude that the Sun's zenith angle played a role in the degree of agreement between the two instruments.

It remains clear that the two instruments have strikingly good agreement for zenith angles greater than 70°, as supported by Fig. 3 and Table 3. Below 70° the correlation dropped rapidly (by almost fifteen percentage points between bins). However, when considering data collected within the SZA range most relevant to stratospheric retrievals (i.e. (85,92.5], see Table 4) the mean percent difference remained below 10 %, with R$^2$ >0.95. From a satellite-validation perspective, this bodes well for

| N | Pandora ($\bar{x}$) | M07 ($\bar{x}$) | Diff. ($\bar{x}$) | Diff. ($\sigma$) | % Diff. | Ratio ($\bar{x}$) | Slope | Intercept | $R^2$ | SZA Range |
|---|---|---|---|---|---|---|---|---|---|---|
| 1135 | 6.488 | 11.648 | -5.160 | 4.00 | 44.3 | 0.602 | 0.226 | 3.860 | 0.128 | (92.5,95.0] |
| 848 | 7.072 | 7.933 | -0.861 | 0.915 | 10.9 | 0.907 | 0.820 | 0.569 | 0.933 | (90.0,92.5] |
| 1208 | 4.470 | 4.767 | -0.297 | 0.463 | 6.2 | 0.955 | 0.866 | 0.340 | 0.955 | (87.5,90.0] |
| 752 | 2.914 | 3.025 | -0.111 | 0.289 | 3.7 | 0.991 | 0.850 | 0.344 | 0.958 | (85.0,87.5] |
| 1660 | 1.895 | 1.855 | 0.040 | 0.199 | 2.2 | 1.076 | 0.841 | 0.336 | 0.960 | (80.0,85.0] |
| 1260 | 1.223 | 1.088 | 0.135 | 0.149 | 12.4 | 1.214 | 0.787 | 0.367 | 0.920 | (75.0,80.0] |
| 235 | 1.003 | 0.848 | 0.155 | 0.129 | 18.3 | 1.289 | 0.737 | 0.378 | 0.899 | (70.0,75.0] |
| 151 | 0.806 | 0.649 | 0.157 | 0.108 | 24.3 | 1.323 | 0.702 | 0.351 | 0.752 | (65.0,70.0] |

**Table 3.** Summary of aggregate statistics for the Pandora/NIWA intercomparison using the standard algorithms and parameters for each instrument. Differences and ratios are relative to the NIWA instrument (i.e. Pandora - M07, Pandora/M07). Statistics were generated using data that were resampled to 5 min means; no further smoothing or binning was applied.

| Statistic | Sunrise | Sunset |
|---|---|---|
| N | 2035 | 2037 |
| SZA Mean | 90.004 | 89.995 |
| SZA Stdev | 0.003 | 0.002 |
| SZA Min | 89.996 | 89.989 |
| SZA Max | 90.010 | 90.000 |

**Table 4.** Solar zenith angle statistics from all (4072) SAGE-III overpasses between 16-March 2017 and 12-August 2017. Solar zenith angles were calculated with respect to a surface instrument's viewing geometry based on the SAGE-III observation time and surface latitude/longitude.

future validation efforts of stratospheric $NO_2$ as >95 % of the inter-instrument variability is accounted for without further correction at zenith angles most relevant to stratospheric observation.

The decrease in correlation at lower SZA's (i.e., as the Sun approaches solar noon) was driven by an apparent offset in the Pandora retrieval at lower SCD's ($\approx$0.55x10$^{16}$ molec cm$^{-2}$) where Pandora seems to lose sensitivity and accuracy, as seen in panels (e-h) of Fig. 3. A similar "tailing" effect due to decreased sensitivity was observed by Knepp et al. (2015) when comparing Pandora $NO_2$ VCD's to in-situ observations and is likely due to the instrument's accuracy limit and light throughput. Therefore, 0.55x10$^{16}$ molec cm$^{-2}$ is considered to be the lower-limit of detection for the current instrument. However, due to the M07's larger slit width and longer focal length, it has more throughput and greater sensitivity than the Pandora, thereby allowing the M07 to continue making reliable measurements up to SZA of 95°.

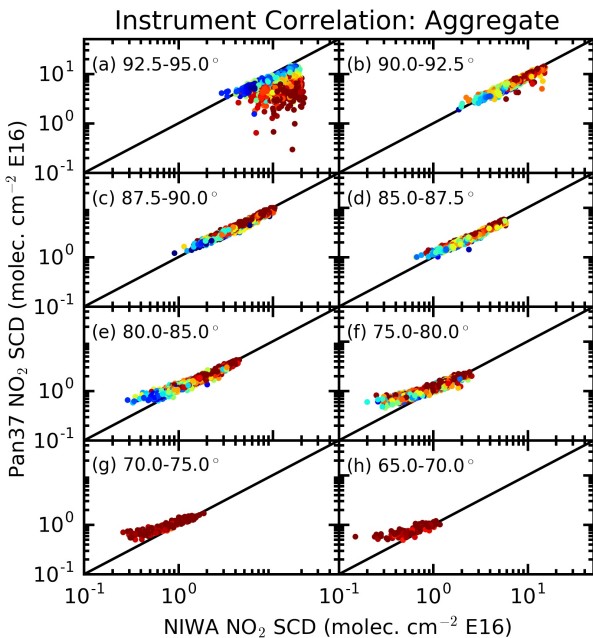

**Figure 3.** Correlation plots for data collected by the Pandora and M07 instruments. Data were resampled to five-minute averages, and color coded according to SZA within the specified bin range (i.e. red colors represent the upper SZA limit, blue represent the lower bounds for each sub-panel). Values within the figure legends indicate SZA ranges.

## 4.2 Seasonal dependence

To better understand seasonal variability seen within the datasets, the data were broken into two major seasons: austral summer (DJFM) and austral winter (JJAS). Seasonal-correlation plots were generated (Fig. 4); they show nearly identical behavior to the aggregate (Fig. 3) with most of the tailing behavior being attributed to winter conditions in agreement with the seasonal cycle depicted in Fig. 2.

SCD and statistical time series plots (Fig. 5) were generated to evaluate the seasonal dependence of both instruments and the inter-instrumental statistics over the year-long operation period. The SCD time series was generated by first binning the data by SZA followed by calculating daily means, which were then smoothed via a 7-day rolling mean. Statistical time series presented in panels (e-af) were generated by resampling each dataset to 5 min averages (i.e., forcing both datasets onto a common date/time index) followed by calculation of the specified statistic on a day-by-day basis, which was then smoothed via a 7-day rolling mean.

Both instruments displayed the expected diurnal (elevated at large SZA, reduced at smaller SZA) and seasonal (elevated $NO_2$ in austral summer (DJFM), followed by reduced levels in winter (JJAS)) trends in $NO_2$ SCD (see also Table 5). This is

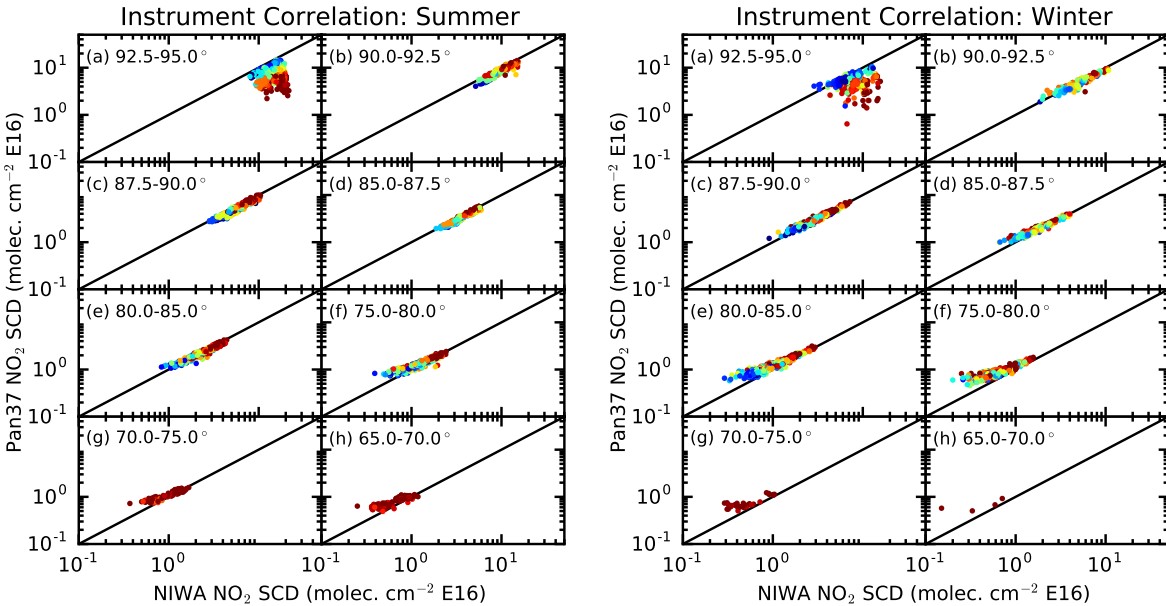

**Figure 4.** Correlation plots for data collected by the Pandora and M07 instruments broken into austral summer/winter. Similar to Fig. 3, data were resampled to five-minute averages, and color coded according to SZA within the specified bin range

in agreement with the expected diurnal behavior (Fishman et al., 2008) and the observed satellite seasonality (Fishman et al. (2008) and Fig. 2).

Inter-instrumental statistics and seasonal dependence were further evaluated. It was observed that the two products tended to have good agreement throughout the year (generally with $\pm10$ %, see Table 5 and Fig. 5 panels (m-p)), with maximal differences at high SZAs (i.e. >2.5° below the horizon, panel (a)) or at very low $NO_2$ (i.e. below the Pandora's sensitivity cutoff, as demonstrated in the tailing behavior of Fig. 3). Further, no seasonal dependence on $R^2$ was observed as $R^2$ remained high throughout the year (>95 %, Table 5 and Fig. 5 panels (m-p)).

Other statistics presented in Table 5 and Figure 5 show a slight seasonal dependence in the measured values. An interesting seasonal and SZA dependence was observed in the ratio and slope data in Table 5 in that the wintertime ratios and slopes were always larger than their summertime counterparts (excluding the pre-sunrise data), and can be most clearly seen in the ratio and difference data in Fig. 5. Ideally, the ratio and inter-instrument offset would remain constant regardless of season, though this was not observed. What is observed is a disproportionate increase in the Pandora-measured SCD (i.e. increasing difference and ratio) from summer to winter compared to M07. Even after removing data where SCD<1x10$^{16}$ the wintertime ratios remain disproportionately high (not shown), therefore this cannot be attributed to Pandora's low-SCD trailing affect. While the source of this seasonal dependence remains unknown, the observed seasonality changed slow enough that the correlations and regressions were not significantly influenced.

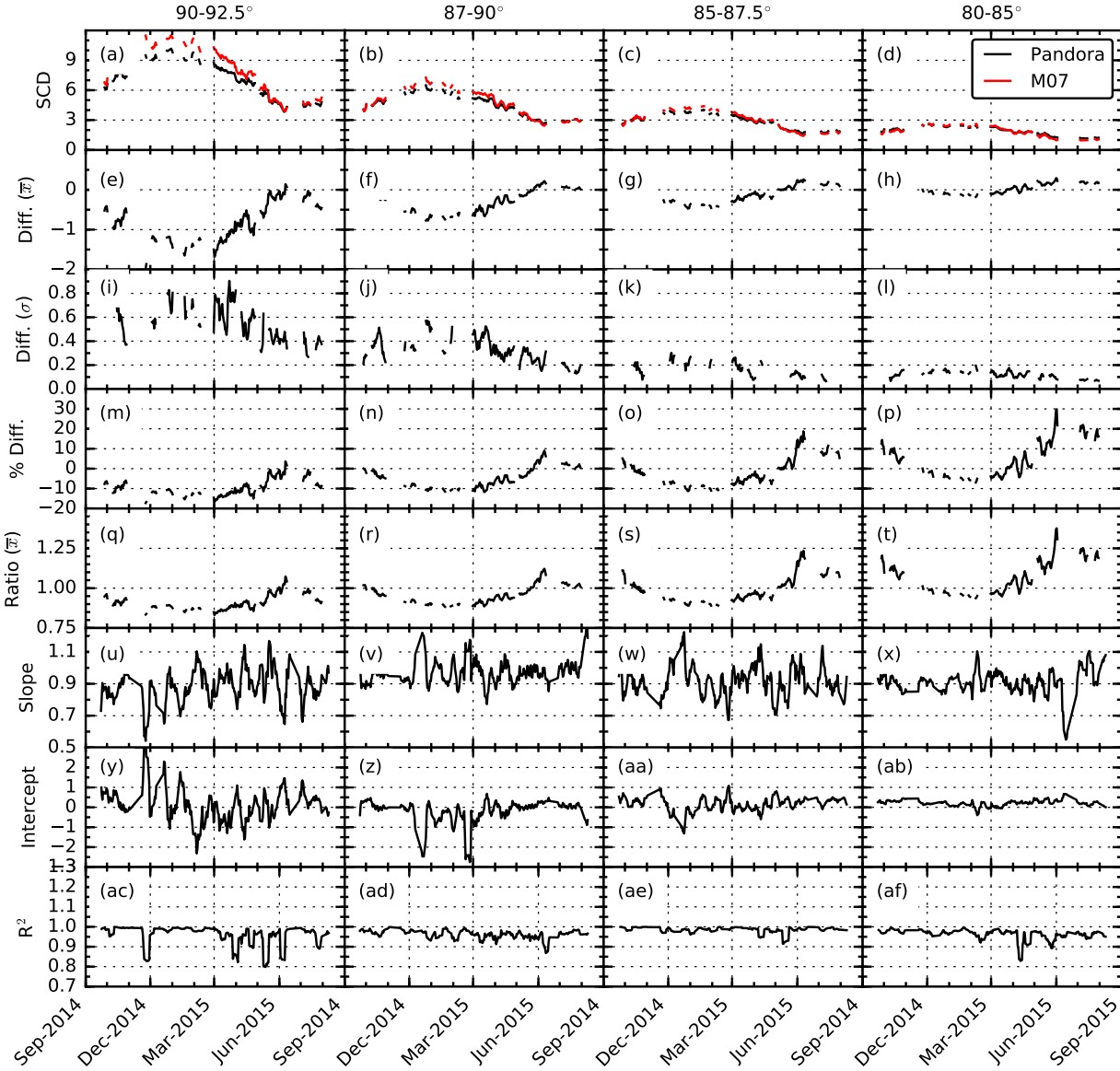

**Figure 5.** Time series for NO$_2$ SCD and daily statistics binned by solar-zenith angle. Data were smoothed by a seven-day rolling mean. Panel descriptions: a-d: SCD for both instruments broken up by SZA; e-h: mean SCD difference (Pandora - M07); i-l: standard deviation of differences; m-p: percent SCD difference; q-t: SCD ratio (Pandora/M07); u-x: line of best fit slope (Pandora vs. M07); y-ab: line of best fit intercept; ac-af: R$^2$ coefficient of correlation. A seven-day normally-weighted rolling mean was applied to smooth the plots and remove higher-frequency fluctuations.

| Season | N | Pandora ($\bar{x}$) | M07 ($\bar{x}$) | Diff. ($\bar{x}$) | Diff. ($\sigma$) | % Diff. | Ratio ($\bar{x}$) | Slope | Intercept | $R^2$ | SZA Range |
|--------|---|---------|---------|----------|----------|---------|----------|-------|-----------|-------|-----------|
| Summer | 694 | 6.957 | 12.735 | -5.778 | 4.056 | -45.4 | 0.582 | 0.181 | 4.651 | 0.071 | (92.5,95.0] |
| Winter | 258 | 4.972 | 8.129 | -3.157 | 2.716 | -38.8 | 0.660 | 0.140 | 3.835 | 0.045 | |
| Summer | 513 | 7.744 | 8.795 | -1.051 | 0.880 | -11.9 | 0.888 | 0.831 | 0.440 | 0.926 | (90.0,92.5] |
| Winter | 202 | 4.839 | 5.124 | -0.285 | 0.577 | -5.6 | 0.961 | 0.829 | 0.593 | 0.917 | |
| Summer | 729 | 4.837 | 5.269 | -0.432 | 0.477 | -8.2 | 0.928 | 0.870 | 0.252 | 0.947 | (87.5,90.0] |
| Winter | 284 | 3.139 | 3.103 | 0.036 | 0.239 | 1.2 | 1.025 | 0.932 | 0.248 | 0.960 | |
| Summer | 469 | 3.138 | 3.340 | -0.202 | 0.271 | -6.1 | 0.952 | 0.870 | 0.233 | 0.952 | (85.0,87.5] |
| Winter | 167 | 1.994 | 1.849 | 0.145 | 0.162 | 7.8 | 1.106 | 0.900 | 0.330 | 0.953 | |
| Summer | 1044 | 2.031 | 2.050 | -0.020 | 0.198 | -1.0 | 1.025 | 0.839 | 0.311 | 0.954 | (80.0,85.0] |
| Winter | 366 | 1.322 | 1.135 | 0.188 | 0.124 | 16.5 | 1.224 | 0.902 | 0.298 | 0.943 | |
| Summer | 872 | 1.254 | 1.146 | 0.108 | 0.146 | 9.5 | 1.170 | 0.774 | 0.368 | 0.925 | (75.0,80.0] |
| Winter | 227 | 0.906 | 0.672 | 0.234 | 0.106 | 34.8 | 1.439 | 0.891 | 0.307 | 0.868 | |
| Summer | 173 | 1.019 | 0.878 | 0.141 | 0.132 | 16.0 | 1.259 | 0.702 | 0.403 | 0.905 | (70.0,75.0] |
| Winter | 36 | 0.736 | 0.521 | 0.215 | 0.104 | 41.3 | 1.516 | 0.782 | 0.328 | 0.775 | |
| Summer | 132 | 0.798 | 0.639 | 0.158 | 0.105 | 24.8 | 1.318 | 0.701 | 0.349 | 0.752 | (65.0,70.0] |
| Winter | 4 | 0.670 | 0.445 | 0.225 | 0.143 | 50.4 | 1.951 | 0.601 | 0.402 | 0.693 | |

**Table 5.** Summary of seasonal statistics for the Pandora/NIWA intercomparison. Similar to Table 3.

## 5 Conclusions

The Pandora instrument was collocated with an NDACC certified instrument (M07 spectrometer) at the NIWA station in Lauder, New Zealand over the period of one year. Spectra from each instrument were processed using separate algorithms to calculate the $NO_2$ SCD throughout the day, but with a focus on twilight periods. We showed that the two instruments and algorithms were well correlated ($R^2 > 0.95$) throughout the entire intercomparison period, and that time of year had minimal impact on the correlation. Further, it was shown that, within a specified SZA bin, a change in SZA influenced the correlation (e.g. Figs. 3 and 4).

The Pandora instrument was shown to have a fundamental limitation due to so-called tailing effects where the instrument seems to lose sensitivity to changes in $NO_2$ slant-column density below $0.55 \times 10^{16}$ molec cm$^{-2}$. The tailing effect is the product of the spectrometer's light throughput, signal-to-noise, and the overall system's precision and accuracy limits. Therefore, Pandora systems may experience sensitivity limitations under extreme-clean conditions. However, the Pandora instrument may prove useful for SAGE-III validations (SZA at time of overpass $\approx 90°$, Table 4). The SAGE-III project plans on deploying Pandora to sites of interest (ideally low-latitude, tropospherically clean environments) with balloon-launching capabilities for

ongoing validation work. Due to Pandora's portability the instrument can also be quickly deployed in response to events of interest (e.g. volcanic eruptions).

## 6 Data availability

All data used within the current study and all code are available from the authors upon request. OMI data are available from
the OMI team via http://avdc.gsfc.nasa.gov/index.php?site=2045907950.

*Competing interests.* The authors declare they have no conflict of interest.

*Acknowledgements.* P. Johnston and R. Querel were supported by NIWA as part of its Government-funded, core research. L. Thomason, D. Flittner, and J. M. Zawodny were supported by NASA's SAGE-III project. T. N. Knepp was supported by NASA's SAGE-III project through the STARS-III contract.

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
