# Peer review of "Intercomparison of Pandora Stratospheric NO2 Slant Column Product with the NDACC-Certified M07 Spectrometer in Lauder, New Zealand"

_Atmospheric Measurement Techniques, 2017_

## Referee Comment (RC1) · Anonymous Referee #2 · 19 Jun 2017

Review of the manuscript 'Intercomparison of Pandora Stratospheric NO2 Slant Column Product with the NIWA M07 NDACC Standard' by Travis N. Knepp et al.

The authors present a comparison of NO2 slant columns measured with a Pandora instrument and the NIWA M07 instrument. Both instruments were employed next to each other at Lauder, New Zealand, and made measurements in zenith sky mode for a whole year (Sep 2014 – Sep 2015). The authors retrieved the measured slant columns using their respective native analysis algorithms and they provide a statistical analysis of the data comparison. The overall motivation for this study is to evaluate the Pandora instrument as a potential SAGE-III validation tool for stratospheric NO2 using

the NDACC approved NIWA instrument as a standard to compare with.

I have no major issues that need to be addressed and a list of specific issues is appended below. However, it would be good to see a bit more of a discussion on where the observed differences between both data sets could potentially be arising from, i.e. although for sure a combination of both, what contributions could come from the actual instrument/raw data and/or could be introduced by the two different processing algorithms. E.g. possible differences in the field of view of the instruments are not discussed nor is discussed if there could potentially be important differences in the actual processing routines used for this intercomparison.

Could one contributing factor to the observed seasonal bias between the data sets be caused by seasonal changes in the cloud cover?

Also a brief statement on how the results can be interpreted in light of the recommendations made by NDACC need to be added.

The manuscript is recommended for publication in AMT.

List of specific comments to be addressed:

Page 1, line 1, Abstract: Add comma: 'In September 2014, a . . ..'

Page 1, line 8, Abstract: Change to '. . . to vertical column density (VCD) is . . .. '

Page 2, line 1: delete 'the' before 'stratospheric'

Page 3, lines 13-18 & page 4, lines 2-10: Please add another paragraph to explain in more detail the processing procedure applied to retrieve the NO2 slant columns discussed in the this paper; the same goes for the NIWA data processing and it would certainly be helpful to have a table summarizing the settings used for both instruments, e.g. wavelength interval used for the fit, which cross-section were used, which polynomial, Ring, has an offset correction been applied, are the NO2 and O3 cross-sections Io corrected? This will help to understand the similarities and differences in the data

processing algorithms applied to each of the data sets.

Page 3, lines 19-21: Can you please provide some evidence that shows that the Pandora instrument used here has a high enough light through-put to make sensible measurements at SZA =>90.

Page 4, line 15: replace ')' with ','

Page 4, line 21: should be 'similarly'

Page 4, lines 26/27: If these are SCDs please state so clearly in the text.

Page 5, line 9: ''. . . the two sites are in phase . . ..'

Page 7, Figure 2: panel (b): any comments re why the displayed ground-based data is clearly higher? Maybe I missed this somehow but this should be included in the caption and also be discussed in the manuscript text. Also state clearly in the first line of the caption if the OMI NO2 data product is SCD or VCD.

Page 8, line 5: '. . . the correlations as seen in Fig. 3'

Page 8, line 6: '. . . where it peaks at . . . Within each panel of Fig. 3 the data are color . . .'

Page 8, line 6: It would also be helpful to refer here to Table 3 as well.

Page 8, line 13: '. . . instruments have strikingly . . .'

Page 9, Tables 3&4: Last column, close with ')' not ']'

Page 10, Figure 3 caption: Change to '(i.e. red colors represent the upper SZA limit, blue colors represent the lower bounds for each individual panel).'

Page 10, line 1: 'The decrease in correlation . . .'

Page 10, line 9: How about something like: 'To better understand the seasonal variability seen within the data sets, the data . . .'

Page 10, lines 10-11: This sentence needs to be improved, change to something like: 'Seasonal correlation plots were generated (Fig. 4); they show nearly identical behaviour to the aggregate (Fig. 3) with most of the tailing behaviour being attributed to winter conditions.'

Page 11, line 11: Change to: '. . . at high SZAs . . . . or at very low NO2 . . .'

Page 11, line 20-22: Can you please address if the observed seasonal dependence can cause issues when using Pandora instruments for satellite validation.

Page 11, line 24: delete '-' between 'NDACC' and 'standard'

Page 11, line 25: replace 'on' with 'using'

Page 11, lines 26-27: Change to: 'We showed that the data obtained using the two instruments and retrieval algorithms were well . . ., and that the time of the year had just minimal impact on the comparison. ' However, didn't you just state in the paragraph above that there actually is a seasonal impact??

Page 11, line 30: The tailing effect should be explained when first mentioned.

Page 12, line1: The SZA range where the Pandora instrument may be useful for SAGE-III validation seems rather limited (around 90 deg, possibly as low as 85 deg) – can you please elaborate a bit on if that is realistic with re to known overpass information for suitable sites.

Page 12, line 2: I don't understand the sentence: 'Lower SZAs may not . . ...'. Can you please explain what you mean here.

———————————————

---

## Referee Comment (RC2) · Anonymous Referee #1 · 18 Jul 2017

General Comments This paper presents an investigation of the Pandora instrument as a means to validate SAGE-III measurements of stratospheric column NO2. The Pandora instrument is a ground-based sun-viewing spectrometer and has proven success in measuring lower tropospheric NO2; Pandoras have been shown to provide reliable measurements of NO2 column amount in areas where the NO2 column is dominated by a polluted troposphere, but their capabilities in areas where the column is dominated by the stratosphere instead have not been previously investigated. Thus, the authors seek to determine the ability of Pandora to observe the stratospheric NO2 column through comparison with a collocated M07 spectrometer (an NDACC-standard instrument) at Lauder, New Zealand, a location considered to be atmospherically clean. The

core motivation for this work is clear and necessary: SAGE-III intends to deliver strato-spheric NO2 column observations as part of its final, end-user data sets, though this quantity is quite difficult to validate. A Pandora would provide a near-ideal ground-based instrument against which to validate SAGE-III measurements, given its potential to retrieve stratospheric NO2 columns and the fact that it is small and mobile, allow-ing it to be set up at many different locations at different times of year for a robust validation effort. This work found good agreement between the M07 instrument and the Pandora, demonstrating the potential usefulness of Pandora to validate SAGE-III observations for solar zenith angles between about 85°-90°. However, paper lackin context necessary for reader to understand full motivation and some other flaws listed here. Therefore, I recommend publication after several major revisions.

Specific Comments – Section 1: More details on the SAGE missions necessary. For instance, please add some more detail about how the SAGE-III/Meteor instrumentation works (including a short description of its viewing geometry, overpass times, etc.), the key SAGE species measurements (besides NO2), and any other data for which SAGE is used. This reviewer is not familiar with this missions, suspects that not all readers will be familiar. Added detail will greatly help to provide context on why validation against Pandora is both necessary and desirable.

What is the citation(s) for the NIWA M07 instrument being considered a standard for stratospheric NO2 measurements? This is unclear.

The NIWA M07 instrument is specifically mentioned only within the last sentence of the introduction; is this the particular instrument that is considered a standard for NO2? Or was it chosen for this intercomparison for another reason, and if so, why? This instru-ment needs to be introduced along with NIWA rather than at the end of the introduction, to prevent confusion over why the M07 instrument was used.

– Section 2.1: "Briefly, the Pandora model used in the current study consisted of..." is unclear; is this different from the "normal" working setup of the instrument, or the

same? A note on this would be helpful. The statement at the end of the section ("...the Pandora only operated in the zenith-observation mode..." also contributes to the lack of clarity.

– Section 3: "...both instruments were operated in their normal states, not in a customized operation mode..." – this gets back to the comment about Section 2.1 about whether Pandora was used the same it has been in previous studies (or not). This statement should be a reiteration of the mode of operation for Pandora (and M07) from Section 2, to make sure it is clear how these instruments were used (and how this does or does not differ from previous studies).

– Section 3: the statistics thing (troposphere beings so different)

– Section 3: Last sentence ("Since Lauder provides a clean, background-level,...") provides a clear statement of the motivation for this work that is not dependent on the specific SAGE mission. This should be perhaps mentioned earlier in the paper (maybe even the introduction after introducing Pandora and Lauder, NZ).

– Section 4: What are the major retrieval uncertainties for Pandora and M07? These should be briefly described, in Section 2 where the two instruments are initially described. Also should make note of any other known limitations/issues related to the instruments or their retrievals.

What does it mean that some datasets were smoothed? Were both Pandora and M07 datasets smoothed, or portions of one or the other instrument's datasets? This statement is unclear. Also, why was five minutes chosen for the averaging time–why not 1 minute, for example?

– Section 4.1: Need to explicitly state that the R2 values are given in Table 3, to make it easier for the reader to find the numbers that support the result that the correlation increased with decreasing SZA. Might even be good to list a few R2 values for some of the SZA bins, since to this reviewer, the correlation for the 87.5-90° SZA bin looks

strongest when looking at the plots in Fig. 3 (though this was not the bin with largest R2). A follow up question is whether the statistical significance of these correlations was tested, to determine if the correlations were statistically different from each other (at least for the bins containing SZAs less than 92.5°); a direct comparison of correlation coefficients can be misleading.

Are R2 values available for the sub-correlations for each panel of Fig. 3? An example for at least one panel might be good, showing how the correlation decreased with lower SZA within that SZA bin (and by extension for the other SZA bins).

Why can the dependence on SZA not be separated from day-to-day chemical variability? I'm not sure what "day to day chemical variability" refers to, so this statement is confusing. Does this refer to the annual variability of the NO2 column, or daily variability of the column? There needs to be a justification for this statement. It would seem that the correlation's dependence on SZA is due to daily photochemistry (available sunlight for photochemical reactions involving NOx), as well as limitations of either instrument at high SZA. So to start the analysis presented in Fig. 3 could be extended, to investigate how the time series of the NO2 columns from both instruments within each SZA bin and over all SZA's compare, comparing to O3 column data, etc.

– Section 4.2: Do the authors have a hypothesis for why the tailing behavior was limited to winter conditions? This would be good to state in the paper.

It's true that the R2 values remained high throughout most of the year, but it can be seen that R2 drops during the winter months for most SZA bins in Fig. 5, such as April-July 2015 bin for the 90-92.5° bin, and for the 80-85° bin. Is this just noise, or is this related to the trends observed in slope and SCD ratio for winter vs. summer? It needs some explanation, and this reviewer is not convinced that it can be said that there is no seasonal dependence seen in the correlation at this time.

– Section 5: The second conclusions paragraph is a little confusing to read. Not quite sure what the message is about, particularly about the twilight retrievals. Some rewording should be all that is needed to make the message clearer.

Technical Corrections: – When referencing parts of a figure, such as panel a in Fig. 3, use parentheses to encapsulate the letter to make it easier to distinguish for the reader (e.g.; Fig. 3 panel a → Fig. panel (a)). – Fig. 4 says "orrelation" in the plot titles rather than "Correlation"

---

## Author Comment (AC1) · 15 Aug 2017

[amtd]copernicus

We appreciate the thorough review from referee #1. The manuscript has been updated to implement the recommendations as described below.

1. Section 1: More details on the SAGE missions necessary. For instance, please add some more detail about how the SAGE-III/Meteor instrumentation works (including a short description of its viewing geometry, overpass times, etc.), the key SAGE species measurements (besides NO2), and any other data for which

[Figure]

SAGE is used. This reviewer is not familiar with this missions, suspects that not all readers will be familiar. Added detail will greatly help to provide context on why validation against Pandora is both necessary and desirable.

(a) More detail was added to introduction.

1. What is the citation(s) for the NIWA M07 instrument being considered a standard for stratospheric NO2 measurements? This is unclear.

   (a) The NIWA group and their instruments have a long heritage of providing data of the highest quality. However, to label this an a "standard" is incorrect. The title and text have been updated to remove confusion in this regard.

1. The NIWA M07 instrument is specifically mentioned only within the last sentence of the introduction; is this the particular instrument that is considered a standard for NO2? Or was it chosen for this intercomparison for another reason, and if so, why? This instrument needs to be introduced along with NIWA rather than at the end of the introduction, to prevent confusion over why the M07 instrument was used.

   (a) Again, the title was updated to remove reference to the NIWA instrument as a community standard. Also, the text was changed to allow introduction of the NIWA instrument under the appropriate section.

1. Section 2.1: "Briefly, the Pandora model used in the current study consisted of. . ." is unclear; is this different from the "normal" working setup of the instrument, or the same? A note on this would be helpful. The statement at the end of the section (". . .the Pandora only operated in the zenith-observation mode. . ." also contributes to the lack of clarity.

[Figure]

(a) The Pandora instruments have been evolving over time, so it is not accurate, at this point, to say there is a standard hardware configuration (e.g. it is not accurate to say they all have the same spectrometer specs etc.). However, there are general characteristics that remain consistent from version to version. Each instrument undergoes the same calibration procedure and the Pandora group have performed multiple intercomparisons between other Pandora units and other DOAS instrumentation (e.g. at the CINDI and CINDI-2 campaigns). Therefore, it is reasonable to trust in the instruments performance. We specify this specific instrument's hardware for clarity.

What is different here is the *mode* of operation. Normally, Pandora instruments track the Sun or look away from the Sun to do elevation scans after the Sun is well above the horizon. In the current study we evaluate the instruments performance in a zenith-only observation mode, specifically during twilight hours. The text has been updated to elucidate this difference in the operation mode.

1. Section 3: ". . .both instruments were operated in their normal states, not in a customized operation mode. . ." - this gets back to the comment about Section 2.1 about whether Pandora was used the same it has been in previous studies (or not). This statement should be a reiteration of the mode of operation for Pandora (and M07) from Section 2, to make sure it is clear how these instruments were used (and how this does or does not differ from previous studies).

(a) As noted above and in the text, the only difference between the mode of operation in the current study and past studies is in the orientation of the entrance optics (i.e. zenith only). The intent of the current study and previous studies is different. Under "normal" Pandora operation the instrument will either do elevation scans, Sun tracking, or some combination of the two. The intention of these normal operation modes is to either collect total VCDs

(Sun tracking) or attempt some vertical profiling (elevation scans). While the elevation scans typically involve a zenith observation, these scans are carried out when the Sun is well above the horizon, not during twilight conditions. Therefore, we cannot provide a comparison with previous studies as the current study is fundamentally different. The current study was carried out to determine whether the Pandora instrument is capable of making twilight observations with the entrance optics oriented in the zenith direction, as stated in the manuscript.

1. Section 3: the statistics thing (troposphere beings so different)

   (a) We are unable to determine what the reviewer means here. No changes made.

1. Section 3: Last sentence ("Since Lauder provides a clean, background-level,. . .") provides a clear statement of the motivation for this work that is not dependent on the specific SAGE mission. This should be perhaps mentioned earlier in the paper (maybe even the introduction after introducing Pandora and Lauder, NZ).

   (a) Given the context, we see manuscript as already meeting the reviewer's request. However, minor wording changes have been implemented within the introduction to bring this out.

1. – Section 4: What are the major retrieval uncertainties for Pandora and M07? These should be briefly described, in Section 2 where the two instruments are initially described. Also should make note of any other known limitations/issues related to the instruments or their retrievals.

   (a) Uncertainties now listed in added section.

1. What does it mean that some datasets were smoothed? Were both Pandora and M07 datasets smoothed, or portions of one or the other instrument's datasets? This statement is unclear. Also, why was five minutes chosen for the averaging time-why not 1 minute, for example?

   (a) We understand the lack of clarity of this statement and appreciate the reviewer bringing this to our attention. When dealing with long time-series data such as the OMI data presented in Fig. 2, the data can appear noisy due to day-to-day fluctuations within the column. This variability can mask trends, and generally make a figure's interpretation difficult. A common technique for bringing out these trends and enhancing a figure's interpretation is to apply a rolling weighted mean, which is what we did in Figs. 2 and 5. Since this "smoothing" was only applied to long time series (i.e. Figs. 2 and 5), this comment was removed from section 4.0. A comment was added to the caption of both Figs. 2 and 5 regarding this smoothing. Again, we appreciate this comment as this would likely have led to confusion of many readers.

   A five-minute block average was applied to both datasets to allow direct comparison of the two data sets. Due to how the data were recorded, the two data sets do not have common time stamps (e.g. Pandora may report an SCD at 12:01:23 while the nearest neighbor for M07 may be at 12:03:11). To temporally align the two datasets a five-minute resampling was performed. This rolling average was done to all data sets and for all analyses. Wording has been changed to clarify this.

1. Section 4.1: Need to explicitly state that the $R^2$ values are given in Table 3, to make it easier for the reader to find the numbers that support the result that the correlation increased with decreasing SZA. Might even be good to list a few $R^2$ values for some of the SZA bins, since to this reviewer, the correlation for the

87.5-90 SZA bin looks strongest when looking at the plots in Fig. 3 (though this was not the bin with largest R2). A follow up question is whether the statistical significance of these correlations was tested, to determine if the correlations were statistically different from each other (at least for the bins containing SZAs less than 92.5); a direct comparison of correlation coefficients can be misleading.

(a) Reference to table added.
    Yes, the R-squared values are significantly different at 95% confidence, except for two cases. Now specified in text.

1. Are R2 values available for the sub-correlations for each panel of Fig. 3? An example for at least one panel might be good, showing how the correlation decreased with lower SZA within that SZA bin (and by extension for the other SZA bins).

(a) No. Due to the nature of the intercomparison and the proposed validation application going to smaller bin ranges is not applicable. The statement within the text "Within each panel of Fig. 3 the data are color coded to correspond to the SZA range within each sub-panel and provide insight into how the short-term change in SZA influenced correlation. As an example, in panel a it is observed that data collected at higher SZA (red-shaded points) were further from the one-to-one line than data collected at lower SZA (blue-shaded points)" was intended to call out the fact that as the SZA changed, individual points moved either closer to or further away from the 1:1 line. The text has been updated to this to be better understood.

1. Why can the dependence on SZA not be separated from day-to-day chemical variability? I'm not sure what "day to day chemical variability" refers to, so this statement is confusing. Does this refer to the annual variability of the NO2 column, or daily variability of the column? There needs to be a justification for

this statement. It would seem that the correlation's dependence on SZA is due to daily photochemistry (available sunlight for photochemical reactions involving NOx), as well as limitations of either instrument at high SZA. So to start the analysis presented in Fig. 3 could be extended, to investi- gate how the time series of the NO2 columns from both instruments within each SZA bin and over all SZA's compare, comparing to O3 column data, etc.

   (a) Agreed. This part of the sentence was confusing and has been deleted.

1. Section 4.2: Do the authors have a hypothesis for why the tailing behavior was limited to winter conditions? This would be good to state in the paper.

   (a) Yes. The "tailing" is only observed at low SCD values. Stratospheric $NO_2$ follows a seasonal cycle shown in Figure 2, with more $NO_2$ in the summer, less in the winter. Therefore, it makes sense that we see lower values in the wintertime. A reference to the seasonal cycle has been added to the text.

1. It's true that the R2 values remained high throughout most of the year, but it can be seen that R2 drops during the winter months for most SZA bins in Fig. 5, such as April-July 2015 bin for the 90-92.5 bin, and for the 80-85 bin. Is this just noise, or is this related to the trends observed in slope and SCD ratio for winter vs. summer? It needs some explanation, and this reviewer is not convinced that it can be said that there is no seasonal dependence seen in the correlation at this time.

   (a) We agree that there is fluctuation within the April-July bin for SZA between 90 and 92.5 degrees. However, this fluctuation is not indicative of a clear seasonal pattern in the coefficients of regression. We support that by failing to see the same behavior in the other SZA bins, and by seeing a similar behavior in December. Further, we would expect to see a similar pattern as

shown in SCD or %diff panels if there were a seasonal dependence. At this time we cannot conclusively state there is a seasonal dependence in the $R^2$ values.

1. – Section 5: The second conclusions paragraph is a little confusing to read. Not quite sure what the message is about, particularly about the twilight retrievals. Some rewording should be all that is needed to make the message clearer.

    (a) Paragraph rewritten.

1. – When referencing parts of a figure, such as panel a in Fig. 3, use parentheses to encapsulate the letter to make it easier to distinguish for the reader (e.g.; Fig. 3 panel a Fig. panel (a)).

    (a) Changes made throughout

1. – Fig. 4 says "orrelation" in the plot titles rather than "Correlation"

    (a) We do not see an error in the titles of figure 4. Perhaps this was an error in the reviewer's file?

---

## Author Comment (AC2) · 15 Aug 2017

[amtd]copernicus

We appreciate the thorough review from referee #2. The manuscript has been updated to implement the recommendations as described below.

1. Page 1, line 1, Abstract: Add comma: 'In September 2014, a ...'

    (a) Comma added.

1. Page 1, line 8, Abstract: Change to '. . . to vertical column density (VCD) is '

(a) "(VCD)" added to text

1. Page 2, line 1: delete 'the' before 'stratospheric'

    (a) Deleted.

1. Page 3, lines 13-18 & page 4, lines 2-10: Please add another paragraph to explain in more detail the processing procedure applied to retrieve the NO2 slant columns dis- cussed in the this paper; the same goes for the NIWA data processing and it would certainly be helpful to have a table summarizing the settings used for both instruments, e.g. wavelength interval used for the fit, which cross section were used, which polynomial, Ring, has an offset correction been applied, are the NO2 and O3 cross-sections Io corrected? This will help to understand the similarities and differences in the data processing algorithms applied to each of the data sets.

    (a) Additional text added, as well as a table containing retrieval information for both instruments and references to retrieval details.

1. Can you please provide some evidence that shows that the Pandora instrument used here has a high enough light through-put to make sensible measurements at SZA $\geq$90.

    (a) This was part of the intention of the work. Prior to this study, there were no studies conducted to conclusively determine whether Pandoras have the requisite light throughput to make twilight observations. Text was added to clarify.

1. Page 4, line 15: replace ')' with ','

(a) Replaced

1. Page 4, line 21: should be 'similarly'

    (a) Changed

1. Page 4, lines 26/27: If these are SCDs please state so clearly in the text.

    (a) When the OMI NO2 product is introduced in the text it is now declared as VCD. The table caption was updated as well to indicate a VCD.

1. Page 5, line 9: '. . . the two sites are in phase . . ..'

    (a) Corrected

1. Page 7, Figure 2: panel (b): any comments re why the displayed ground-based data is clearly higher? Maybe I missed this somehow but this should be included in the caption and also be discussed in the manuscript text. Also state clearly in the first line of the caption if the OMI NO2 data product is SCD or VCD.

    (a) As mentioned in the text, we have not inverted the surface instruments' SCD to a VCD. Figure 2, panel b presents the OMI VCD and the Pandora/M07 SCD. The three instruments are *not* plotted here to show quantitative agreement, rather it is to demonstrate that the surface instruments are in phase with the OMI seasonal cycle for stratospheric NO2. Clarification has been added to the text and the figure's caption.

1. Page 8, line 5: '. . . the correlations as seen in Fig. 3'

    (a) Corrected

1. Page 8, line 6: '. . . where it peaks at . . . Within each panel of Fig. 3 the data are color ... '

   (a) All changes implemented.

1. Page 8, line 6: It would also be helpful to refer here to Table 3 as well.

   (a) Reference to table added.

1. Page 8, line 13: '. . . instruments have strikingly . . .'

   (a) Verb tense changed

1. Page 9, Tables 3&4: Last column, close with ')' not ']'

   (a) It is common mathematical notation to use parentheses to indicate an exclusive range, and square brackets to indicate an inclusive range. Therefore, we chose to use square brackets to indicate the ranges specified in tables 3 & 4 are inclusive of the high SZA.

1. Page 10, Figure 3 caption: Change to '(i.e. red colors represent the upper SZA limit, blue colors represent the lower bounds for each individual panel).'

   (a) Hyphenation removed

1. Page 10, line 1: 'The decrease in correlation . . .'

   (a) Change implemented

1. Page 10, line 9: How about something like: 'To better understand the seasonal variability seen within the data sets, the data . . .'

(a) Your wording provides better understanding of what we are doing. Change implemented.

1. Page 10, lines 10-11: This sentence needs to be improved, change to something like: 'Seasonal correlation plots were generated (Fig. 4); they show nearly identical behaviour to the aggregate (Fig. 3) with most of the tailing behaviour being attributed to winter conditions.'

   (a) Again, your wording allows quicker understanding of the text. Change implemented.

1. Page 11, line 11: Change to: '. . . at high SZAs . . .. or at very low NO2 . . .'

   (a) Change implemented.

1. Page 11, line 20-22: Can you please address if the observed seasonal dependence can cause issues when using Pandora instruments for satellite validation.

   (a) Seasonal variability in satellite validation is a well known issue. Unfortunately, we cannot determine the seasonal dependence when comparing either Pandora or M07 to SAGE-III until after the observations are made. We can be certain that there will be a seasonal dependence on how well Pandora and M07 agree with SAGE-III, but that seasonality will have to be accounted for at a later time.

1. Page 11, line 24: delete '-' between 'NDACC' and 'standard'

   (a) Hyphen deleted.

1. Page 11, line 25: replace 'on' with 'using'

    (a) Change implemented.

1. Page 11, lines 26-27: Change to: 'We showed that the data obtained using the two instruments and retrieval algorithms were well . . ., and that the time of the year had just minimal impact on the comparison. ' However, didn't you just state in the paragraph above that there actually is a seasonal impact??

    (a) You are correct. It is clearly demonstrated within the manuscript that the seasonal impact was +/- 10%. The sentence should have said "minimal impact on correlation". Now corrected.

1. Page 11, line 30: The tailing effect should be explained when first mentioned.

    (a) Tailing effect now defined in the conclusions section.

1. Page 12, line1: The SZA range where the Pandora instrument may be useful for SAGE-III validation seems rather limited (around 90 deg, possibly as low as 85 deg) - can you please elaborate a bit on if that is realistic with re to known overpass information for suitable sites.

    (a) A table was added to show SZA statistics for all SAGE-III/ISS observations collected thus far. The SZA were calculated with respect to a potential surface instrument's viewing geometry. The table shows the SZA to be tightly grouped about 90 degrees.

1. Page 12, line 2: I don't understand the sentence: 'Lower SZAs may not . . ...'. Can you please explain what you mean here.

    (a) That sentence was superfluous and has been removed.